# Screening of Characteristic Metabolites in Bee Pollen from Different Floral Sources Based on High-Resolution Mass Spectrometry

**DOI:** 10.3390/foods14244305

**Published:** 2025-12-14

**Authors:** Lanhua Liu, Zhiwei Sun, Aiyuan Liang, Run Zhang, Siqi He, Yaling He, Min Zhang, Xingjiang Li, Xuefeng Wu

**Affiliations:** 1Instrumental Analysis Center, Hefei University of Technology, Hefei 230009, China; 2Anhui Fermented Food Engineering Research Center, Key Laboratory for Agricultural Products Processing of Anhui Province, School of Food and Biological Engineering, Hefei University of Technology, Hefei 230601, China

**Keywords:** bee pollen, different floral sources, LC-HRMS, GC-MS, metabolites

## Abstract

Bee pollen is a natural nutrient substance collected by bees from plants. Its metabolites have been extensively studied, yet the characteristic metabolites of bee pollen from different floral sources have not been clearly identified. In this study, we collected four types of bee pollen (tea, rose, rapeseed, and corn pollen) from across China and analyzed their volatile and non-volatile metabolites using liquid chromatography-high-resolution mass spectrometry (LC-HRMS) and gas chromatography-mass spectrometry (GC-MS). At the same time, the nutritional substances (Including polyphenols, organic acids, and sugars) were precisely quantified. The results showed that the total phenols (5 mg GAE/g) and total flavonoids (0.27 mg RE/g) content of corn pollen were significantly higher (*p* < 0.05) than those of other pollens, and the contents of polyphenols such as naringenin were relatively high, indicating strong antioxidant potential. Rose pollen was rich in protein (0.04 g/g) and flavonoid glycosides. Tea pollen was prominent in the content of polyphenol glycosides and amino acid derivatives, while rapeseed pollen performed well in phenolic acids (Ferulic acid), as well as specific sugar (Mannose). We identified the differential metabolites of these bee pollen through orthogonal partial least squares discriminant analysis (OPLS-DA) (VIP > 1). It was also stipulated that metabolites with a VIP value greater than 1.5 showed significant differences and could be used as characteristic metabolites for differentiating pollen (*p* < 0.05). The representative metabolites of bee pollen were as follows: rapeseed pollen—ferulic acid; tea pollen—malic acid; corn pollen—epicatechin; and rose pollen—fumaric acid. This study provides a research basis for evaluating the quality, traceability, and metabolite exploration of bee pollen.

## 1. Introduction

Bee pollen, a natural nutrient-rich substance collected by bees from various plants, is abundant in proteins, vitamins, lipids, polyphenols, and other bioactive compounds [1]. Its exceptional nutritional value has rendered it a highly regarded food of significant interest. Research has confirmed that bee pollen exhibits anti-inflammatory, antioxidant, and gastrointestinal protective effects [2]. Although the metabolites of bee pollen have been extensively studied, technological advancements have revealed that numerous metabolites still require further exploration [3]. Furthermore, the metabolites of bee pollen from different floral sources showed significant differences, which might be attributed to factors such as geographical conditions. Through ^1^H-NMR (Nuclear magnetic resonance) and HPLC analyses of bee pollen samples from Spain, China, Australia, and other places, it was found that there were significant differences in the contents of sucrose, nucleotides (Such as adenosine, uridine, cytidine), amino acids, and flavanols among the samples from different countries [4]. However, the specific metabolic markers for bee pollen from different floral sources have not yet been clearly identified.

Liquid chromatography-high resolution mass spectrometry (LC-HRMS) and gas chromatography-mass spectrometry (GC-MS) are two fundamental technologies for metabolite detection [5]. GC-MS is particularly well-suited for analyzing small to medium-sized molecules that are volatile and thermally stable, facilitating efficient identification with the aid of standard spectral libraries [6]. In contrast, LC-HRMS can achieve efficient separation of non-volatile metabolites and precise determination of accurate molecular weight [7]. Currently, LC-HRMS has been widely applied in detecting various metabolites such as phospholipids, polyphenols, and small molecules in food samples [8,9]. Meanwhile, GC-MS has also been employed for the analysis and identification of differential flavor markers [10]. Therefore, based on the analytical advantages of GC-MS and LC-HRMS for metabolites, combining the two methods can effectively conduct a comprehensive analysis of the volatile and non-volatile metabolites in bee pollen.

This study aims to conduct a comparison of the four representative Chinese bee pollen samples (tea, rose, rapeseed, and corn pollen). To systematically identify and clearly distinguish the characteristic metabolites of bee pollen from different floral sources. This study provides a theoretical basis for the quality control, traceability, and functional utilization of bee pollen.

## 2. Materials and Methods

### 2.1. Materials and Chemical Reagents

Bee pollen, including tea, rose, rapeseed, and corn pollen, was provided by Anhui Fengxian Bee Industry Co., Ltd. (Anqing, China). All samples were collected from beekeepers across various regions of China (Figure 1A). They were transported under light-protected conditions and stored in sterile containers at −20 °C. For bee pollen, a five-point sampling method was employed, with five replicates of 500 g collected from each type. None of the samples had been stored for more than three months before analysis.

Both organic acid standards (Oxalic acid, tartaric acid, quinic acid, malic acid, lactic acid and fumaric acid, purity ≥ 98%) and sugar standards (Rhamnose, arabinose, mannose, sucrose, maltose and fucose, purity ≥ 98%) were from Shanghai Yuanye Biotechnology Co., Ltd (Shanghai, China). Polyphenol standards (Arbutin, protocatechuic acid, epigallocatechin, hydrous catechin, chlorogenic acid, caffeic acid, epigallocatechin gallate, epicatechin, p-coumaric acid, ferulic acid, rutin, epicatechin gallate, taxifolin, astragalin, liquiritigenin, ellagic acid, quercetin, naringenin, apigenin, kaempferol, isorhamnetin, baicalein, isoliquiritigenin, pinocembrin and ursolic acid, purity ≥ 98%) from Beijing Solarbio Science and Technology Co., Ltd (Beijing, China). Methanol, formic acid, acetonitrile, and ethanol were all analytical grade, from Sinopharm Chemical Reagent Co., Ltd (Shanghai, China).

### 2.2. Basic Physical and Chemical Index Determination

#### 2.2.1. Determination of Basic Physicochemical Indicators

Total phenol and flavonoid contents were determined using modified Folin–Ciocalteu and spectrophotometric methods, respectively [11]. Quantification was performed using gallic acid and catechin calibration curves, and the results are presented as gallic acid equivalents and catechin equivalents, respectively.

Protein content in bee pollen was determined by the Kjeldahl method [12]. Samples were digested into a solution using a graphite digestion instrument from Hanon Advanced Technology Group Co., Ltd (Jinan, China). Ammonium in the sample was then released by alkalization, absorbed by boric acid solution, and titrated with a standard hydrochloric acid solution to calculate the protein content.

#### 2.2.2. Determination of Polyphenol Content

Polyphenol content in bee pollen was determined by LC-HRMS [13]. Bee pollen (1.0 g) was weighed into a 50 mL centrifuge tube, and 20 mL of 70% ethanol aqueous solution (pH = 2, adjusted with HCl) was added. The mixture was homogenized at 1000 rpm for 5 min. Subsequently, the mixture was ultrasonically extracted at room temperature with 100 W for 30 min. The bee pollen solution (1.0 mL) was loaded onto an activated HLB solid-phase extraction cartridge (3.0 mL of methanol and 3.0 mL of distilled water were added for activation), allowed to flow naturally, dried with nitrogen, rinsed with 1.0 mL ultrapure water, dried again with nitrogen, and then eluted with 1.0 mL methanol. The eluate was then dried with nitrogen and reconstituted by thoroughly shaking with 200 μL of methanol solution. The supernatant was transferred to a brown autosampler vial and analyzed by Liquid Chromatography-Quadrupole Electrostatic Field Orbitrap Mass Spectrometer (LC-QE Orbitrap).

Analysis was performed using an Eclipse Plus C18 column (2.1 × 100 mm, 1.8 μm, Agilent) at a flow rate of 0.15 mL/min, the column temperature was 40 °C, and the sample temperature was 4 °C. Mobile phase A was ultrapure water (0.1% formic acid), and mobile phase B was acetonitrile. The gradient elution program was as follows: 0 min, 5% B; 15 min, 30% B; 22 min, 90% B; 27 min, 90% B; 28 min, 5% B; 30 min, 95% B. Mass spectrometry conditions were as follows: Electrospray Ionization (ESI) spray voltage was 3200 V. The instrument parameters were set as follows: evaporation temperature at 350 °C, ion transfer temperature at 500 °C, sheath gas at 35 arb, and auxiliary gas at 10 arb. Analysis was carried out in negative ion mode with a scan range of 50–750 *m*/*z*. Following the establishment of standard curves relating concentration to peak area, the polyphenol content in the samples was quantified. Each sample was tested in triplicate to guarantee measurement consistency.

#### 2.2.3. Determination of Sugar Content

Sugar content in bee pollen was determined by LC-HRMS [14]. Bee pollen samples (3.0 g) were weighed into a centrifuge tube, and 20 mL of ultrapure water was added. The mixture was homogenized at 1000 rpm for 5 min. Subsequently, ultrasonic extraction was performed at room temperature with 100 W for 30 min. 1 mL of the supernatant was filtered through a 0.22 μm polyethersulfone filter membrane into an autosampler vial for analysis. Analysis was performed using an ACQUITY BEH Amide Column (2.1 mm × 150 mm, 1.7 μm, Waters, Milford, MA, USA). Mobile phase A was a 0.1% ammonia aqueous solution, and mobile phase B was a 0.1% ammonia in acetonitrile solution. The gradient elution program was: 0 min, 78% B; 10 min, 55% B; 10.1 min, 78% B. The flow rate was 0.2 mL/min, the injection volume was 1.0 μL, and the column temperature was 60 °C. Mass spectrometry conditions were the same as Section 2.2.2.

#### 2.2.4. Determination of Organic Acid Content

Organic acid was determined by LC-HRMS [15]. Bee pollen (5.0 g) was weighed into a 50 mL centrifuge tube, and 20 mL of ultrapure water was added. The mixture was homogenized at 1000 rpm for 5 min, followed by ultrasonic extraction at 100 W at room temperature for 30 min. Then, 20 mL of absolute ethanol was added and allowed to stand for 10 min. The mixture was centrifuged at 10,000 rpm for 5 min. The supernatant was collected and made up to 50 mL in a volumetric flask. And then the sample solution (1 mL) was filtered through a 0.22 μm nylon filter membrane into an autosampler vial for injection. Separation was performed using a ZORBAX Eclipse Plus C18 Column (5 μm, 4.6 × 250 mm, Agilent, Santa Clara, CA, USA). The mobile phase was a mixture of methanol and water—methanol/water (0.1% formic acid (*w*/*w*)) ratio was 3:97—at a flow rate of 0.8 mL/min. The column temperature was 30 °C, and the injection volume was 10 μL. Mass spectrometry conditions were the same as Section 2.2.2.

#### 2.2.5. Determination of Volatile Metabolites in Bee Pollen

Volatile metabolites were analyzed using Gas Chromatography-Mass Spectrometry (GC-MS) [16]. Sample (3.0 g) was weighed and dissolved in 10 mL of ultrapure water, then homogenized for 5 min. Subsequently, 1 mL of the sample solution was loaded onto an activated HLB solid-phase extraction column, allowed to flow naturally (same as in Section 2.2.2). The eluate was dried with nitrogen, redissolved in 150 μL methanol, and 50 μL of 100 mmol/L 2-methyl-3-heptanone was added as an internal standard for online detection.

A HP-5MS Column (30 m × 0.25 mm, 0.25 µm, Agilent) was used for analysis. The temperature program was as follows: initial temperature of 40 °C (held for 2 min), then temperature ramped to 250 °C at 10 °C min^−1^. The injector temperature was 250 °C, the injection volume was 0.2 μL, the split ratio was 10:1, and the carrier gas was helium at a flow rate of 1 mL/min. The Electron Ionization (EI) source voltage was 70 eV, temperature was 230 °C, ionization electron multiplier voltage was 1682 V, emission current was 34.6 μA, interface temperature was 230 °C, scan range was 20–500 *m*/*z*, and solvent delay was 4 min. The identification of volatile compounds was performed by comparing their mass spectra with those in the NIST 20 Standard Spectral Library using a computer-based search. Compounds exhibiting a match quality exceeding 80% were selected for further analysis. The retention index (RI) for each volatile was calculated using a homologous series of n-alkanes and confirmed by comparing with both mass spectral data and relevant literature. For semi-quantification, 2-octanol was employed as an internal standard, and the relative peak areas of the compounds were compared.

#### 2.2.6. Determination of Non-Volatile Metabolites in Bee Pollen

Metabolites in bee pollen were analyzed using LC-HRMS [17,18]. A sample of bee pollen weighing 1.0 g was dissolved in 3 mL of ultrapure water and subsequently diluted with 10 mL of methanol. The resulting mixture was centrifuged at 10,000 rpm for 10 min at room temperature. The supernatant was then filtered through a 0.22 μm nylon membrane and injected into an autosampler vial for detection. Analysis was performed using a ZORBAX Eclipse Plus C18 Column (2.1 × 100 mm, 1.8 μm, Agilent). The flow rate was 0.35 mL/min, the column temperature was 40 °C, and the injection temperature was 4 °C. Mobile phase A was a 0.1% formic acid aqueous solution, and mobile phase B was acetonitrile. The gradient elution program was: 0 min, 5% B; 15 min, 30% B; 22 min, 90% B; 27 min, 90% B; 28 min, 5% B; 30 min, 95% B. Mass spectrometry conditions were the same as Section 2.2.2. Determination of Polyphenol Content with simultaneous positive and negative ion scanning. The original mass spectrum file is subjected to noise reduction, peak extraction, retention time correction, and alignment. Subsequently, candidate metabolites are screened and annotated based on information such as accurate mass (Δmass < 5 ppm), typical fragment ions (diagnostic fragments), and neutral loss characteristics (neutral loss). Through matching scores with standard samples’ retention time and MS/MS fragments (ChemSpider), as well as the inferred structure based on accurate mass and fragment characteristics from the public database (mzCloud).

### 2.3. Data Processing

Statistical analyses were conducted using SPSS Statistics software (version 26). Data for each group are presented as mean ± standard deviation (SD). All statistical comparisons employed Duncan’s test, with a significance level set at *p* < 0.05. Principal Component Analysis (PCA) and Orthogonal Partial Least Squares Discriminant Analysis (OPLS-DA) were used to analyze the metabolites of bee pollen from different floral sources. In this analysis, metabolites with a VIP value greater than 1 were defined as differential substances among bee pollen from different floral sources (Sugars, organic acids, polyphenols, volatile metabolites, and non-volatile metabolites). These analyses were performed using MetaboAnalyst (https://www.metaboanalyst.ca/, accessed on 20 April 2025) to generate corresponding plots. Heatmaps and VIP value plots were generated using Chiplot (https://www.chiplot.online/, accessed on 25 April 2025).

## 3. Results and Discussion

### 3.1. Basic Indicators of Bee Pollen from Different Floral Sources

The pollen from different floral sources showed significant differences in total phenols, total flavonoids, and protein content. This is due to the differences in the distribution of these sources. Regional climate and soil conditions have a significant impact on the formation of nutritional components in bee pollen [19,20]. In terms of total phenol content, the contents of corn pollen and tea pollen were both relatively high (both 5 mg/g), significantly higher than that of rapeseed pollen (*p* < 0.05) (Figure 1B). Tea pollen possessed an intermediate total phenol content, lower than corn but higher than rapeseed pollen, suggesting a potential antioxidant advantage. The total flavonoid content of corn pollen was the highest (0.27 mg RE/g), followed by that of rapeseed pollen (0.21 mg RE/g), while the contents were similar but lower in tea pollen and rose pollen (Figure 1C). Due to the significant biological activities such as anti-inflammatory, free radical scavenging, and cardiovascular protection exhibited by flavonoids, corn pollen may have greater application potential in the development of functional foods [21,22]. The protein content was significantly superior in rose pollen, registering 0.04 g/g. Tea pollen contained an intermediate level (0.022 g/g), whereas rapeseed and corn pollen possessed relatively lower protein contents (Figure 1D). Protein, as an important nutritional component of bee pollen, plays a crucial role in human amino acid supplementation and immune regulation; thus, rose pollen can be considered a representative source of high-protein bee pollen [23,24].

Overall, the four types of bee pollen from different floral sources showed significant differences in total phenol, total flavonoid, and protein contents. These differences may be closely related to the plant source of the pollen, environmental conditions, and the foraging habits of bee species [20,25]. Therefore, further investigation into the specific metabolite differences among bee pollen from different floral sources is needed.

### 3.2. Metabolite of Bee Pollen from Different Floral Sources

We conducted an exploration of the volatile and non-volatile metabolites in bee pollen and carried out qualitative and quantitative analysis of some of the nutrients (Organic acids, sugars, and polyphenols) within it. In addition, the recovery rate of polyphenols was also determined (Appendix A). In tea pollen, the markedly higher levels of oxalic acid (45.42 μg/g) and malic acid (99.15 μg/g), compared with rapeseed and rose pollen, suggest that these two acids dominate its organic acid profile (Table 1). Rose pollen, which contained abundant tartaric acid (4.23 μg/g) and citric acid (5.93 μg/g), might possess a stronger sour flavor perception, which has been reported to affect pollinator taste sensitivity and thus potentially influence foraging decisions [26,27]. Corn pollen exhibited relatively high levels of oxalic acid and malic acid (48.08 μg/g and 70.23 μg/g), while its lower tartaric and citric acid content indicated a milder acidic profile [26].

Tea pollen contained significantly higher amounts of epigallocatechin (1090.98 μg/g), epicatechin (1254.67 μg/g), and p-coumaric acid (1051.24 μg/g) than other pollen, revealing strong potential for antioxidant activity. Polyphenols such as epicatechin and epigallocatechin are known to neutralize reactive oxygen species (ROS) through hydrogen atom donation and metal-chelating mechanisms, which may enhance the physiological resilience of bees consuming such pollen [28]. Likewise, p-coumaric acid has been reported to upregulate detoxification pathways in honeybees, thereby supporting immune function and mitigating pesticide-induced oxidative stress [29]. These biochemical activities suggest that tea pollen may provide superior protective value for insect consumers.

Rose pollen, rich in mannose (8910.53 μg/g) and arbutin (524.52 μg/g), showed potential for functional food applications linked to whitening and antioxidant activity [30,31]. The presence of arbutin and flavonoid glycosides such as astragalin (675.22 μg/g) and taxifolin (14.52 μg/g) may further contribute to redox-modulating properties, benefiting pollinators by reducing cellular oxidative damage [32,33]. Rapeseed pollen contains high levels of isorhamnetin (4190.68 μg/g) and isoliquiritigenin (18.46 μg/g), compounds associated with anti-inflammatory and cardiovascular protective effects [34]. Corn pollen is similarly enriched in isorhamnetin (7454.73 μg/g), which may serve as a biochemical marker for its anti-inflammatory potential.

Beyond their biochemical roles, these metabolites may influence bee foraging behavior. Bees have been shown to prefer pollen not only for its protein content but also for the presence of specific bioactive compounds that support colony health [35]. For instance, pollen rich in antioxidant flavonoids (Such as epicatechin, rutin, taxifolin) may enhance larval development and adult immunity, making such pollen sources more attractive during foraging. Thus, the metabolite profiles identified in this study not only reflect chemical diversity but also highlight important nutritional and ecological implications for pollinators.

A total of 53 volatile metabolites and 169 non-volatile metabolites were identified in these bee pollen samples, and significant differences were observed (Appendix A). The pollen showed distinct differences in amino acid derivatives, phenolic derivatives, and fatty acid metabolites (Figure 2A,B). These differences in metabolite distribution can serve as important chemical fingerprints, which can be used to trace the source of pollen and conduct quality assessment. Corn pollen exhibited higher abundance in various fatty acid metabolites, rose pollen was prominent in phenolic acids, while tea pollen had high contents of amino acid derivatives, consistent with its high protein and free amino acid content. Overall, corn pollen was superior in total flavonoids, certain polyphenols, and fatty acid metabolites, making it suitable for developing antioxidant and anti-inflammatory functional products [36]. Rose pollen had high contents of protein and specific flavonoid glycoside, making it a potential raw material for high-nutrition and skin-care functional products. Tea pollen was characterized by high polyphenol glycoside conjugates and amino acid derivatives, possessing potential antioxidant and health-promoting values [19]. Rapeseed pollen, on the other hand, had advantages in specific phenolic acids (Isorhamnetin and ferulic acid) and sugar (Mannose), which may make it suitable for the development of cardiovascular health products. Future research could combine bioactivity assays and human intervention trials to further validate their functional efficacy and metabolic stability.

### 3.3. Differential Metabolite of Bee Pollen from Different Floral Sources

PCA showed that the chemical composition, volatile metabolites, and non-volatile metabolites of the four pollen sources showed a significant separation trend (Figure 3). Specifically, chemical components in tea pollen and corn pollen were far apart in the PC1 and PC2 directions, indicating significant differences in their overall compositional makeup (Figure 3A). Rose pollen and rapeseed pollen were relatively concentrated, suggesting higher similarity in their basic chemical component profiles. The volatile metabolites of each pollen formed distinct clusters in the PC1 direction, with rose pollen distributed separately within a narrow range, indicating higher stability in its volatile component composition (Figure 3B). Rapeseed pollen and corn pollen showed partial overlap in the PC2 direction, suggesting that volatile metabolites in bee pollen showed certain similarities. The non-volatile metabolites of pollens were completely separated, especially tea pollen and rapeseed pollen, which showed significant differences in the PC1 direction, reflecting that non-volatile metabolites are important chemical fingerprints for distinguishing floral sources (Figure 3C). The PLS-DA results (Figure 4A–C) further reinforced the PCA separation trend: the chemical component clusters of bee pollen from different floral sources had clear boundaries, and the classification model exhibited high discriminative power. The clustering boundaries for volatile metabolites of tea pollen and rose pollen tightened, indicating higher consistency in their volatile metabolite composition (Figure 4B). All pollen non-volatile metabolites showed significant separation in the direction of the first principal component, suggesting that non-volatile compounds have high explanatory power (>30%) in identifying pollen sources (Figure 4C).

The differential metabolites in bee pollen from different floral sources were identified using a VIP value cutoff of 1. This analysis revealed that hydroxyphenylacetic acid, protocatechuic acid, and ursolic acid served as critical discriminators for floral origin. Among these, protocatechuic acid and ursolic acid were highest in tea pollen, potentially conferring strong antioxidant activity [21]. Among volatile metabolites, β-ionone, nonanal, and 2-methylheptane were relatively enriched in rose pollen and rapeseed pollen, which may be closely related to their floral fragrance characteristics [25]. Among non-volatile metabolites, phenylalanine, p-hydroxycinnamic acid, and quercetin-3-O-glucoside were prominent in corn pollen, suggesting its potential functional advantages in anti-inflammatory, antioxidant, and cardiovascular protection [37].

The reliability of the OPLS-DA model is demonstrated by the Model validation parameters (Appendix A), with its Accuracy value, R^2^, and Q^2^ all approaching 1. These parameters collectively indicate that the OPLS-DA model we constructed is robust, effective, and highly statistically significant. Consequently, the differential metabolites subsequently identified based on the VIP values of this model can accurately reflect the biological differences between groups.

### 3.4. Characteristic Metabolites of Bee Pollen from Different Floral Sources

Metabolites with a VIP value greater than 1.5 were defined as differential representative metabolites. Those metabolites with high content and whose content in a specific pollen was significantly (*p* < 0.05) higher than that in other bee pollens were defined as characteristic metabolites of that pollen. The results illustrated the main origins and characteristic metabolites of bee pollen from different floral sources (Figure 5). Among them, ferulic acid, protocatechuic acid, and caffeic acid were characteristic metabolites of rapeseed bee pollen. Theophylline, malic acid, and acetanilide were characteristic metabolites of tea pollen. Epicatechin, fructose-arginine, and propyl acetate were characteristic metabolites of corn pollen, and leucine, sucrose, and fumaric acid were characteristic metabolites of rose pollen.

## 4. Conclusions

This study was based on high-resolution mass spectrometry and employed targeted/non-targeted metabolomics methods to systematically compare the metabolite profiles of four types of bee pollen (tea, rose, rapeseed, and corn pollen). A total of 6 kinds of organic acids, 6 sugars, 25 polyphenols, 53 volatile metabolites, and 169 non-volatile metabolites were identified. The total phenols, flavonoids, and naringin content in corn pollen were significantly higher than those in the other bee pollen. It demonstrated excellent antioxidant potential. Rose pollen was rich in flavonoid glycosides and proteins, tea pollen was rich in polyphenol glycosides and amino acid derivatives, and rapeseed pollen had significantly higher phenolic acids than the other bee pollen. Principal component analysis (PCA) and partial least squares discriminant analysis (PLS-DA) indicated significant differences in metabolites among the pollen (*p* < 0.05). Metabolites with VIP > 1.5 could significantly distinguish the bee pollen from different floral sources, and malic acid (tea pollen), fumaric acid (rose pollen), ferulic acid (rapeseed pollen), and epicatechin (corn pollen) were identified as characteristic metabolites of the bee pollen. These findings provide a comprehensive metabolic fingerprint of bee pollen, facilitating source tracing and quality control, and offering targeted guidance for the development of functional foods. Future research can focus on verifying its biological activity in vivo and optimizing processing techniques to retain key metabolites. This work enriches our understanding of the nutritional diversity of bee pollen and lays a theoretical foundation for its industrial application.

## Figures and Tables

**Figure 1 foods-14-04305-f001:**
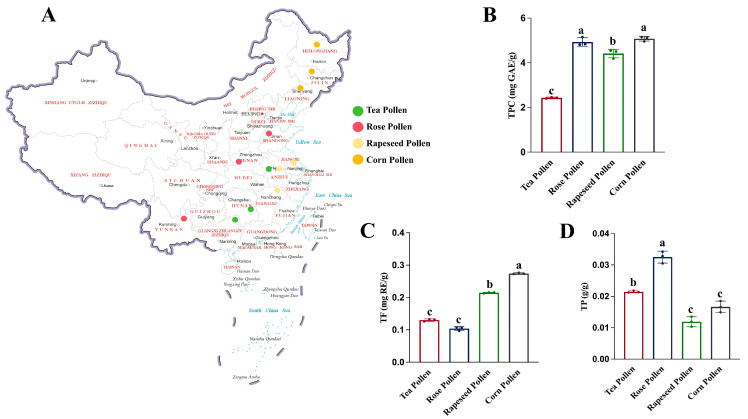
Geographical distribution and basic physicochemical indicators of bee pollen from different floral sources. Note: Main collection and production areas (**A**), Total phenol content-TPC (**B**), Total flavonoid content-TF (**C**), Total protein content-TP (**D**). Different letters (a-c) in the figures indicated significant differences (*p* < 0.05)

**Figure 2 foods-14-04305-f002:**
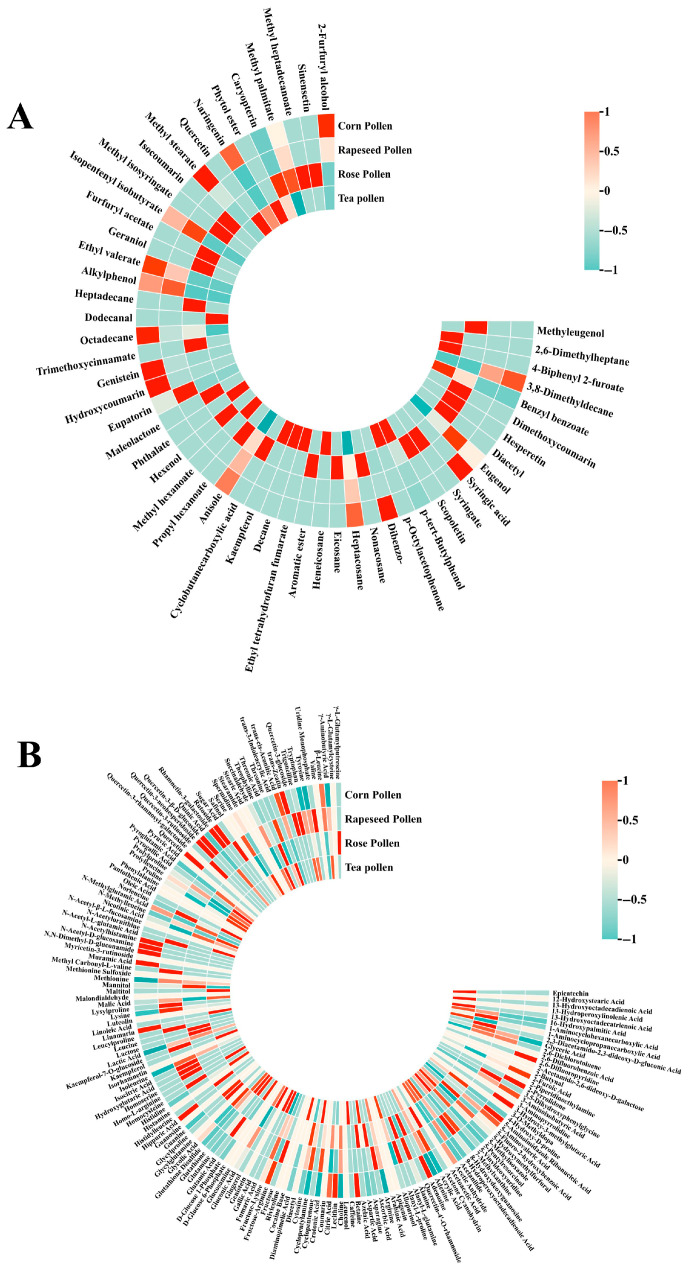
Metabolites of bee pollen from different floral sources. Note: Volatile metabolites (**A**), Non-volatile metabolites (**B**). The scale (−1 to 1) represents the normalized relative abundance of metabolites (z-score standardized), with 1 indicating the highest and −1 the lowest abundance across samples.

**Figure 3 foods-14-04305-f003:**
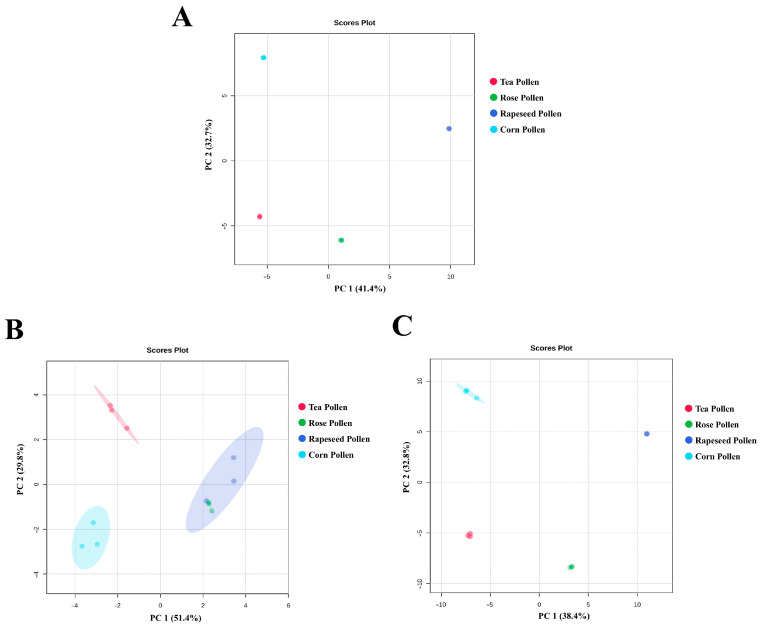
PCA score plot of pollen metabolites from different floral sources. Note: Chemical components (**A**), Volatile metabolites (**B**), Non-volatile metabolites (**C**).

**Figure 4 foods-14-04305-f004:**
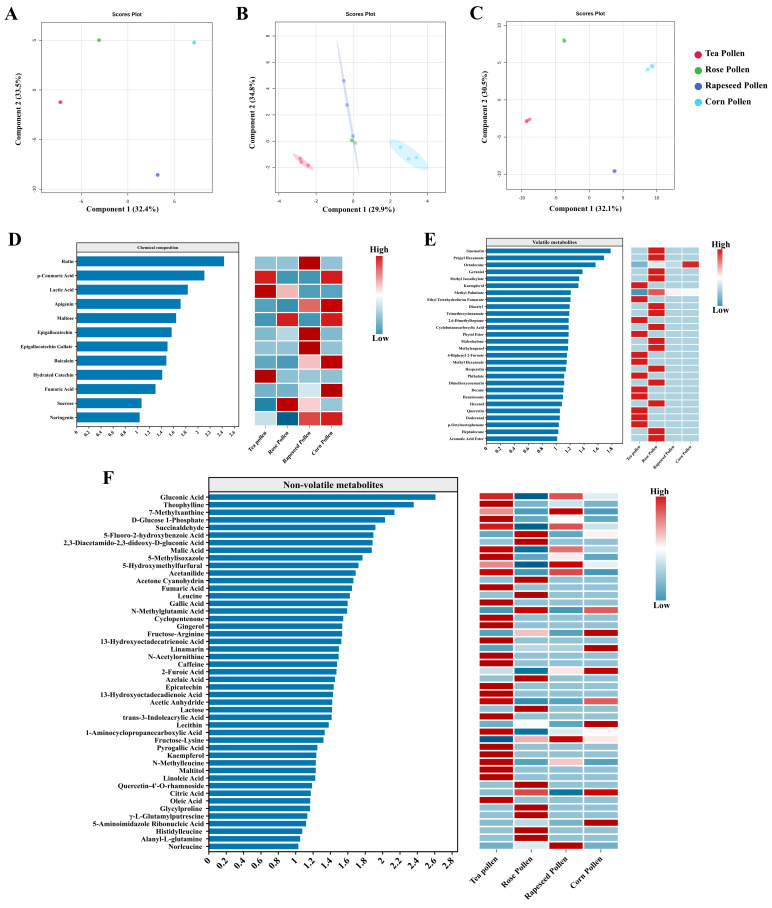
PLS-DA score plots and VIP value plots (VIP > 1.0) of bee pollen metabolites from different floral sources. Note: Chemical component score plot (**A**), Volatile metabolite score plot (**B**), Non-volatile metabolite score plot (**C**), Chemical component VIP value plot (**D**), Volatile metabolite VIP value plot (**E**), Non-volatile metabolite VIP value plot (**F**).

**Figure 5 foods-14-04305-f005:**
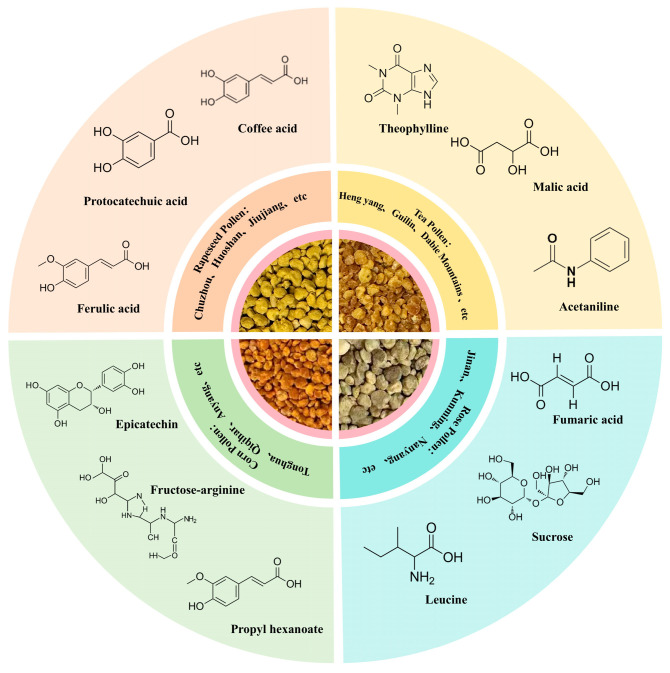
Representative metabolites of bee pollen from different floral sources.

**Table 1 foods-14-04305-t001:** Content of organic acids, sugars, and polyphenols in bee pollen from different floral sources.

Category	Compound	Tea Pollen	Rose Pollen	Rapeseed Pollen	Corn Pollen
Organic acids (μg/g)	Oxalic acid	45.4 ± 2.97 a	27.93 ± 2.67 b	20.09 ± 1.26 c	48.08 ± 2.44 a
	Tartaric acid	0.56 ± 0.03 c	4.23 ± 1.15 a	0.33 ± 0.05 c	1.04 ± 0.16 b
	Quinic acid	1.82 ± 0.21 c	2.63 ± 0.42 a	0.19 ± 0.02 d	2.27 ± 0.32 b
	Malic acid	99.15 ± 0.93 a	50.31 ± 1.05 b	0.19 ± 0.03 d	2.27 ± 0.31 c
	Lactic acid	—	—	5.93 ± 1.07 b	30.05 ± 3.91 a
	Fumaric acid	1.04 ± 0.02 a	0.94 ± 0.12 a	—	—
Sugars (μg/g)	Rhamnose	182.64 ± 0.42 a	—	—	181.13 ± 0.2 a
	Arabinose	—	—	319.32 ± 30.59 a	—
	Mannose	4754.18 ± 21.54 c	8910.53 ± 28.36 a	6869.97 ± 85.54 b	5720.17 ± 37.76 b
	Sucrose	—	124.24 ± 0.87 a	—	123.03 ± 8.45 a
	Maltose	—	454 ± 6.67 a	269.33 ± 2.09 b	159.33 ± 6.51 c
	Fucose	1470.24 ± 0.31 a	1470.07 ± 0.12 a	1469.99 ± 0.02 a	1470.15 ± 0.23 a
Polyphenols (μg/g)	Arbutin	325.81 ± 13.51 c	524.45 ± 21.34 a	323.07 ± 11.02 c	407.61 ± 21.39 b
	Protocatechuic acid	—	6.55 ± 0.34 b	52.3 ± 4.47 a	—
	Epigallocatechin	1090.98 ± 55.61 a	—	29.59 ± 2.26 b	—
	Hydrous catechin	2.17 ± 0.39 a	1.42 ± 0.15 b	—	—
	Chlorogenic acid	—	2.23 ± 0.95 b	3.62 ± 0.80 a	—
	Caffeic acid	—	—	22.11 ± 1.34 a	—
	Epigallocatechin gallate	31.47 ± 0.22 a	—	—	—
	Epicatechin	1254.67 ± 86.16 a	—	—	1254.67 ± 28.16 a
	p-Coumaric acid	1051.2 ± 46.44 a	1012.36 ± 50.97 a	497.53 ± 24.56 b	—
	Ferulic acid	—	—	271.56 ± 15.27 a	—
	Rutin	244.21 ± 16.85 a	214.21 ± 3.05 b	—	—
	Epicatechin gallate	2.05 ± 0.23 a	0.05 ± 0.01 b	0.54 ± 0.07 b	0.16 ± 0.02 c
	Taxifolin	1.64 ± 0.19 c	14.52 ± 1.35 b	131.93 ± 4.37 a	0.34 ± 0.26 d
	Astragalin	268.87 ± 24.34 b	675.2 ± 23.4 a	114.55 ± 7.13 c	641.8 ± 13.65 a
	Liquiritigenin	—	—	12.19 ± 1.35 a	—
Polyphenols (μg/g)	Ellagic acid	3243.03 ± 103.56 b	96.65 ± 5.12 c	6255.26 ± 20.83 a	6772.29 ± 18.77 a
	Quercetin	3215.07 ± 98.24 b	110.4 ± 5.46 c	6033.4 ± 19.05 a	6515.18 ± 16.03 a
	Naringenin	83.58 ± 4.59 c	—	957.72 ± 25.53 b	1252.17 ± 41.23 a
	Apigenin	87.59 ± 4.64 a	58.77 ± 3.41 b	35.89 ± 1.41 c	—
	Kaempferol	5617.08 ± 14.06 a	2851.69 ± 75.57 b	408.82 ± 16.41 c	2089.33 ± 54.12 b
	Isorhamnetin	1388.35 ± 35.14 c	1025.2 ± 22.53 c	4190.68 ± 44.11 b	7454.73 ± 18.26 a
	Baicalein	—	—	—	38.7 ± 1.74 a
	Isoliquiritigenin	—	—	18.46 ± 2.23 a	—
	Pinocembrin	13.74 ± 2.95 c	17.39 ± 1.2 b	54.18 ± 3.13 a	10.51 ± 1.45 c
	Ursolic acid	83.63 ± 2.67 b	231.93 ± 2.25 a	50.66 ± 1.91 d	64.07 ± 2.48 c

Different letters (a-d) in the same row indicate significant differences (*p* < 0.05); — indicates not detected.

## Data Availability

The original contributions presented in the study are included in the article/Appendix A; further inquiries can be directed to the corresponding author.

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
