# Peer review of "Screening of Characteristic Metabolites in Bee Pollen from Different Floral Sources Based on High-Resolution Mass Spectrometry"

_foods, 2025, doi:10.3390/foods14244305_

Round 1
Reviewer 1 Report
Comments and Suggestions for Authors
The manuscript entitled "Analysis and Screening of Metabolite Profiles of Different 2
Flower-Sourced Bee Pollen Based on High-Resolution Mass 3
Spectrometry" deals with a very interesting topic. The experiment is well designed and different analytical methods were used, which gives additional value to the paper.
However, some there are few points that need to be clarified:
- It is obvious that Introduction section is missing. Authors should write this part, and provide all the relevant data concerning background of the topic as well as aim of the research
- For the Determination of Polyphenol Content SPE procedure was conducted. Are there any data concerning recovery values of the polyphenols analyzed?
- Figure 2 should be clarified, what does the scale from -1 to 1 represents?
Reviewer 2 Report
Comments and Suggestions for Authors
The manuscript entitled "Analysis and Screening of Metabolite Profiles of Different Flower-Sourced Bee Pollen Based on High-Resolution Mass Spectrometry" has been submitted to Foods journal. The authors chose 4 types of bee pollen and performed targeted and non targeted analysis using modern high resolution instruments.
There is no introduction section, at the beginning of the manuscript one may find "materials and methods section" which is quite weird.
The aim of the study is clear, well defined and the selection of methods is justified.
There are significant differences in the volatile and non-volatile metabolite profiles of the four pollens, however this is not the first attempt to characterize bee pollen in China, which is the largest producer of honey.
What is definitely missing in this study and at the same time very interesting for the reader is the pesticide content, which is known to be present in the pollen from different regions. Using untargeted analysis and scanning from 50-750 m/z the authors must have seen some xenobiotics. Do you have these data?
The figures have very small font and therefore are illegible.
Conclusion section should be extended.
Reviewer 3 Report
Comments and Suggestions for Authors
Although the experimental work in this project is interesting, the manuscript lacks several elements necessary for publication in a scientific journal.
To begin with, it lacks an introduction that mentions the importance of this research and its background.
The experimental data are not well presented. (lines 49-53, please, Place the purity of the standard and the brand. How did you create the calibration curves to quantify each of the flavonoids reported in Table 1?)
The discussion of results is incomplete.
The conclusions need to be more coherent.
The work is interesting, it just needs a better presentation.
Reviewer 4 Report
Comments and Suggestions for Authors
The manuscript presents a systematic comparative study on bee pollen from four floral sources (tea, rose, rapeseed, and corn pollen), focusing on their physicochemical characteristics, metabolic composition, and discriminative metabolite profiling using LC-HRMS and GC-MS techniques. The topic is highly relevant to current research trends in functional food chemistry, metabolomics, and apiculture product authentication.
The study’s aim, to correlate metabolic diversity with botanical origin and potential functional value, is scientifically sound. The combination of advanced analytical platforms (LC-HRMS, GC-MS) and multivariate statistical tools (PCA, OPLS-DA) represents a modern and appropriate strategy for compositional differentiation of natural matrices such as bee pollen.
However, while the manuscript demonstrates solid experimental work and potentially meaningful findings, its presentation, organization, and analytical interpretation require significant improvement to meet the standards of an international scientific journal.
Major Comments
- The introduction should more clearly define the novelty of the study relative to existing literature on bee pollen metabolomics (e.g., studies from 2020–2025 using LC-MS-based fingerprinting).
- The manuscript would benefit from a concise summary of known metabolic differences among pollen types and how this study extends beyond prior compositional analyses.
Suggest strengthening the introduction by citing recent metabolomic or nutritional profiling studies of bee pollen. - Although the analytical procedures are described in detail, the structure is overly procedural and difficult to follow.
- Some critical information is missing or scattered:
- Replicate numbers for each analysis;
- Calibration ranges and validation parameters (linearity, R², LOD/LOQ);
- Internal standards used for LC-HRMS quantification.
Suggest condensing methods into clear subsections emphasizing experimental design, validation, and data processing strategy, rather than step-by-step procedural details.
- Results mention concentrations such as “total phenols (5 mg/g)” and “total flavonoids (0.27 mg/g),” but do not specify units of expression consistency (e.g., mg GAE/g DW).
- The manuscript should include error margins (mean ± SD, n=3) and statistical indicators (P-values) for all quantitative data.
Suggest presenting all compositional results in tables with uniform units, clear statistical representation, and consistent sample identifiers. - LC-HRMS and GC-MS analyses are core to the paper’s novelty, but spectral identification criteria (mass accuracy, MS/MS matching, library used) are not described.
Suggest to detail compound identification confidence levels (e.g., Level 1–3 according to Metabolomics Standards Initiative) and provide representative chromatograms or extracted ion chromatograms in Supplementary Data. - PCA and OPLS-DA are mentioned, but the manuscript does not discuss model validation parameters (e.g., R²X, Q², permutation tests).
Suggest adding a statistics validation subsection summarizing model performance and discussing the biological interpretation of clustering patterns and discriminant metabolites. - The discussion focuses mainly on numerical differences (e.g., “corn pollen showed higher polyphenols”) but lacks biochemical or ecological interpretation.
Suggest expanding the discussion to relate identified metabolites (e.g., ferulic acid, naringenin, epicatechin) to antioxidant mechanisms, pollen nutritional quality, or bee foraging behavior. - The manuscript would benefit from comprehensive English language editing to correct minor grammatical inconsistencies and enhance scientific precision (e.g., “increased to 250 ℃ at a rate of 10 ℃/min” → “temperature ramped to 250 °C at 10 °C min⁻¹”).
Reviewer 5 Report
Comments and Suggestions for Authors
The work presented in this manuscript is interesting for the characterization and differentiation of bee pollen types. Appropriate analytical techniques are applied, and the conclusions are consistent with the results obtained. However, there are important aspects of the manuscript that, in my opinion, should be revised and corrected. There is no an introduction paragraph. It has to be included explaining the background, interest and objetives of the work that has been carried out. The content shown in Figures 1, 2, 3, and 4 is difficult to visualize, and if possible, their quality should be improved. The chemical formulas included in Figure 5 should be reviewed. For example, the name "acid coffee" is included, when I assume it should be caffeic acid. Similarly, the formulas for sucrose and propyl hexanoate should be reviewed. According to the instructions for authors, references should be listed including all authors, with the first spell of the last name capitalized followed by the initials of the first name. The journal title should be abbreviated in italics, followed by the year in bold, the volume, and the page range.
In the following other comments are included.
Line 21. All types of the determined compounds should be included in the abstract, avoiding the use of etc.
Line 39. Gas chromatography – mass spectrometry could be included as a keyword.
Line 43. Paragraph Materials and instruments. The number of samples of each type of bee pollen sample is important to have an idea of the variability among samples of the same type.
Line 156. Could authors confirm the gradient elution program? The final part of the program seems odd to me. (27 min, 90% B; 28 min, 5% B; 30 min, 95% B)
Line 174. Samples of camellia pollen are mentioned here. It was probably inserted improperly.
Line 178. Comment of the values of total phenol content do not fit with those appearing in Figure 1B. Also revise comment of values of protein content of rose pollen in Figure 1D.
Lines 208 – 211. Comment of the values of concentration of citric acid should be revised according to those appearing in Table 1.
Table 1. Data of amino acids and fatty acids are included in this table, but the methods of analysis of these chemical parameters are not described and the standards used are not mentioned in paragraph 1.1. Where data of these compounds come from? This should be explained.
Round 2
Reviewer 3 Report
Comments and Suggestions for Authors
THANKS FOR THE CORRECTIONS, IT LOOKS BETTER NOW.
I would like you to please add the purity of each of the standards used in the analysis.
Author Response
I would like you to please add the purity of each of the standards used in the analysis.
Thank you very much for your detailed suggestions. We have made the corresponding revisions as follows:
Both organic acid standards (Oxalic acid, tartaric acid, quinic acid, malic acid, lactic acid and fumaric acid, purity ≥ 98%) and sugar standards (Rhamnose, arabinose, mannose, sucrose, maltose and fucose, purity ≥ 98%) were from Shanghai Yuanye Biotechnology Co., Ltd. Polyphenol standards (Arbutin, protocatechuic acid, epigallocatechin, hydrous catechin, chlorogenic acid, caffeic acid, epigallocatechin gallate, epicatechin, p-coumaric acid, ferulic acid, rutin, epicatechin gallate, taxifolin, astragalin, liquiritigenin, ellagic acid, quercetin, naringenin, apigenin, kaempferol, isorhamnetin, baicalein, isoliquiritigenin, pinocembrin and ursolic acid, purity ≥ 98%) from Beijing Solarbio Science & Technology Co., Ltd. Methanol, formic acid, acetonitrile, and ethanol were all analytical grade, from Sinopharm Chemical Reagent Co., Ltd.
Reviewer 4 Report
Comments and Suggestions for Authors
Agree with the revised manuscript.
Author Response
Thank you for your suggestions on the article
Reviewer 5 Report
Comments and Suggestions for Authors
In the new version the authors have considered the points raised in the first version, improving the quality of the manuscript.
Nevertheless, I consider some revision, mainly text editing, is still necessary.
In their response to the reviewer's questions, authors have provided answer to the considered queries raised about this work but, for some reason, they have not included some of them in the revised version of the manuscript. In the following those points are included (numbers refer to the sections of the document of responses to the reviewer's questions). 3-4.- Revise Figure 5; 5.- Prepare list of references according to the instructions to authors; 8.- To include gas chromatography - mass spectrometry as a keyword; 11.- Use tea pollen instead of camellia pollen; 13.- Comment about content of citric acid. Authors should consider these points to prepare the final version of the manuscript.
Author Response
In their response to the reviewer's questions, authors have provided answer to the consideredaueries raised about this work but, for some reason, thev have not included some of them in therevised version of the manuscript. In the following those points are included (numbers refer to thesections of the document of responses to the reviewer's questions). 3-4.- Revise Figure 5; 5..Prepare list of references according to the instructions to authors; 8.- To include gaschromatography - mass spectrometry as a keyword; 11.- Use tea pollen instead of camelliapollen; 13.- Comment about content of citric acid. Authors should consider these points toprepare the final version of the manuscript.
We are extremely grateful for your valuable suggestions on how to clarify the uniqueness of this research compared to the recent studies on bee pollen metabolomics. We have made the necessary revisions as per your instructions. Due to the scattered locations being quite far apart, we have made annotations in the revised version.